

# Phosphate sorption and desorption by two contrasting volcanic soils of equatorial Africa

Sara Gonzalez-Rodriguez[1] and Maria Luisa Fernandez-Marcos[1,2]

[1] Department of Soil Science and Agricultural Chemistry, Universidad de Santiago de Compostela, Lugo, Spain
[2] Institute of Agricultural Biodiversity and Rural Development, University of Santiago de Compostela, Lugo, Spain

## ABSTRACT

Volcanic soils cover 1% of the Earth's surface but support 10% of the world's population. They are among the most fertile soils in the world, due to their excellent physical properties and richness in available nutrients. The major limiting factor for plant growth in volcanic soils is phosphate fixation, which is mainly attributable to active species of aluminium and iron. The sorption and desorption of phosphate is studied on the surface horizons of two African agricultural soils, a silandic Andosol (Rwanda) and a vitric Andosol (São Tomé and Principe). Both soils are slightly acid. The silandic Andosol is rich in active aluminium forms, while the vitric Andosol has high amounts of crystalline iron and aluminium oxides. Sorption isotherms were determined by equilibrating at 293K soil samples with phosphate solutions of concentrations between 0 and 100 mg P L$^{-1}$ in NaNO$_3$; phosphate was determined by visible spectrophotometry in the equilibrium solution. To study desorption, the soil samples from the sorption experiment were equilibrated with 0.02 M NaNO$_3$. The isotherms were adjusted to mathematical models. In almost all the concentration range, the adsorption of phosphate by the silandic Andosol was greater than 90% of the amount added, being lower in the vitric Andosol but always higher than 65%. The high sorption by the silandic Andosol is attributed to its richness in non-crystalline Fe and Al, while in the vitric Andosol crystalline iron species seem to play a relevant role in the adsorption. The sorption isotherms of both soils fitted to the Temkin model, the adjustment to the Langmuir or Freundlich models being unsatisfactory; throughout the range studied, the sorption increases with increasing phosphorus concentration, a maximum sorption is not predictable (as occurs when the sorption is adjusted to the Langmuir model). For an added P concentration of 100 mg L$^{-1}$ (3.2 mmol L$^{-1}$), the sorption is 47.7 µmol P g$^{-1}$ in the silandic Andosol and 41.6 µmol P g$^{-1}$ in the vitric Andosol. The desorption is low and the comparison of the sorption and desorption isotherms reveals a pronounced hysteresis, that is, the irreversibility of the sorption. The high phosphate sorption and its irreversibility are comparable to those published for other volcanic soils with high contents of allophane, active aluminium and free iron. The strong phosphate adsorption is a serious limiting factor for plant growth, which requires a careful management of phosphorus fertilization.

Corresponding author
Maria Luisa Fernandez-Marcos,
mluisa.fernandez@usc.es

# INTRODUCTION

Volcanic soils cover 1% of the Earth's surface but support 10% of the world's population (*Neall, 2009*). These soils are among the most fertile in the world, due to their excellent physical properties and their richness in available nutrients. From the chemical point of view, their major limitation is phosphate fixation, attributable mainly to active species of aluminium and iron (*Nanzyo, 2002*).

Adsorption of oxyanions by soils and mineral surfaces has received considerable attention (*Adegoke et al., 2013*; *Gasparatos et al., 2006*; *Jiang et al., 2015*; *Kumar et al., 2016*; *Parfitt, 1979*), due to their role as nutrients or pollutants. Mineral surfaces, particularly those of metal (hydr)oxides, can adsorb oxyanions by specific and non-specific complexation mechanisms (*Goldberg & Johnston, 2001*). Specifically adsorbed oxyanions are strongly bound to the surface through covalent bonds formed by ligand exchange with surface OH groups (inner sphere complexes). Non-specifically adsorbed oxyanions bind weakly to the surface by electrostatic attraction through an interposed water molecule (outer sphere complexes).

Soil available phosphorus is constituted by phosphorus in the soil solution plus the so-called *labile phosphorus*, which easily passes from the soil solid phase into solution. The specific adsorption of phosphate by active soil surfaces transforms it into non-labile, hence non-available, phosphorus.

The presence of active forms of aluminium and iron, such as oxides, oxyhydroxides, short-range order silicates (allophane, imogolite), Al(Fe)-humus complexes, is a characteristic of volcanic soils. These materials confer the soil the ability to adsorb phosphate. Phosphate fixation by iron and aluminium oxides or allophane is a limiting factor for plant growth, given the condition of essential nutrient of phosphorus (*Sanchez, Palm & Buol, 2003*). The high phosphate fixation by iron and aluminium oxides can be shown by a value of iron extractable by dithionite-citrate higher than 4% and leads to phosphate fertilization needs of more than 100 mg P kg$^{-1}$. The high phosphate fixation by allophane or imogolite can be evidenced by a value of pH in NaF higher than 10 and leads to phosphate fertilization needs of more than 200 mg P kg$^{-1}$.

In Rwanda, the most densely populated country in Africa, volcanic soils occupy an area of 700 km$^2$, being crucial for population livelihoods (*Neall, 2009*). São Tomé (São Tomé and Principe), with an area of 857 km$^2$, is a small volcanic island, mainly basaltic in nature, near the West African coast, which is part of the volcanic alignment known as the Cameroon Hot Line (*Deruelle, Ngounouno & Demaiffe, 2007*).

Volcanic soils have been extensively studied in the world, particularly with respect to phosphorus adsorption (*Van Ranst et al., 2004*; *Auxtero, Madeira & Sousa, 2008*; *Hashimoto et al., 2012*; *Valle et al., 2015*), but there is a dearth of information on volcanic soils in Africa. A number of published papers address the phosphate sorption by African soils (*Sibanda & Young, 1986*; *Loganathan, Isirimah & Nwachuku, 1987*; *Frossard et al., 1992*; *Adepoju, 1993*; *Arduino et al., 1993*; *Adetunji, 1997*; *Henry & Smith, 2003*; *Nwoke et al., 2003*; *Gichangi, Mnkeni & Muchaonyerwa, 2008*). A few papers deal with phosphate adsorption by African volcanic soils: *Duffera & Robarge (1999)* studied P sorption by

four highland soils of Ethiopia, among which an Andosol developed from volcanic ash. *Gimsing, Szilas & Borggaard (2007)* reported the sorption of phosphate and glyphosate by four variable-charge Tanzanian soils, including an Andosol. *Siewe et al. (2008)* studied the phosphate adsorption by an Andosol of the Bambouto Mountains (Cameroon). *Woumfo, Siewe & Njopwouo (2015)* investigated the usefulness of an Andosol from Bambouto Mountain (Cameroon) as an adsorbent to remove phosphorus from polluted waters. As far as we know, there are no published studies on P desorption in African soils.

Phosphate sorption capacity is a principal factor regulating P concentration in the soil solution and hence available phosphorus. Understanding the P sorption and release characteristics of soils is important in determining the fate of applied P fertilizer. Correcting P deficiencies by fertilizer application must take into account the P fixation characteristics of the soil (*Duffera & Robarge, 1999*). Phosphorus sorption by soils must be considered not only to determine the P fertilizing requirement but also the management of phosphate fertilizers. This is of paramount importance in developing countries, where the price of fertilizers often restricts the possibility of fertilizing agricultural fields.

In order to contribute to a better management of phosphorus fertilization in African volcanic soils under a changing environment, this paper aims to study the phosphate sorption and desorption by the surface horizons of two agricultural soils of equatorial Africa, a silandic Andosol (*IUSS Working Group WRB, 2015*) on volcanic ash (Rwanda) and a vitric Andosol on basaltic material (São Tomé and Principe). Silandic Andosols are characterized by their richness in short-range-order silicates (allophane, imogolite), while vitric Andosols are characterized by the presence of volcanic glass and a lower content of short-range-order minerals (lesser degree of weathering).

## MATERIALS AND METHODS

### Soils

The surface horizons (0–20 cm) of two agricultural soils located in Rwanda and São Tomé and Príncipe were used in the study. The first one, developed from volcanic ash, classifies as silandic Andosol (*IUSS Working Group WRB, 2015*), while the soil of São Tomé Island classifies as vitric Andosol. The geographic location and some relevant properties of the two studied soils are presented in Table 1. The aluminium extracted by acid oxalate ($Al_o$) estimates the ''active aluminium'' (aluminium in organic complexes, in non-crystalline hydrated oxides, in allophane and imogolite) (*Garcia-Rodeja et al., 2004*; *Parfitt & Childs, 1988*). The iron extracted by acid oxalate ($Fe_o$) corresponds essentially to ferrihydrite (*Nanzyo, 2002*; *Parfitt & Childs, 1988*). The concentrations of aluminium and iron extractable by acid oxalate fulfil the requirement for andic properties ($Al_o + \frac{1}{2} Fe_o$ >2%; *IUSS Working Group WRB, 2015*) in the soil of Rwanda and for vitric properties ($Al_o + \frac{1}{2} Fe_o$: 0.4–2%) in the soil of São Tomé.

The dithionite-citrate extracts the total free iron ($Fe_d$), including non-crystalline forms of iron ($Fe_o$) and crystalline oxyhydroxides (goethite, hematite, magnetite ...) (*García-Rodeja et al., 2007*; *Parfitt & Childs, 1988*). The aluminium extracted by dithionite-citrate ($Al_d$) includes non-crystalline aluminium forms (although it is ineffective in the extraction

**Table 1** Location, pH, organic carbon, texture, available P, aluminium and iron extracted by acid oxalate and by dithionite-citrate and pH in NaF of the studied soils.

| Soil | Location | Altitude (m) | pH | C (%) | Texture | Mehlich-3 P (mg kg$^{-1}$) | Fe$_o$ (%) | Al$_o$ (%) | Fe$_d$ (%) | Al$_d$ (%) | pH$_{NaF}$ |
|------|----------|--------------|-----|-------|---------|----------------------------|------------|------------|------------|------------|------------|
| **Silandic Andosol** | 01°36′32″S, 29°32′57″E | 311 | 5.60 | 6.18 | Sandy loam | 32.2 | 3.46 | 2.48 | 4.60 | 1.81 | 11.07 |
| **Vitric Andosol** | 00°20′00″N, 06°39′00″E | 2,368 | 6.30 | 7.46 | Silty loam | 13.5 | 1.11 | 0.40 | 7.84 | 1.74 | 8.39 |

Notes.

Fe$_o$, Al$_o$: Fe and Al extracted by acid ammonium oxalate (*Blakemore, 1983*).

Fe$_d$, Al$_d$: Fe and Al extracted by dithionite-citrate (*Holmgren, 1967*).

of non-crystalline aluminosilicates) as well as aluminium occluded in crystalline iron oxyhydroxides (*García-Rodeja et al., 2007*). The concentration of Fe$_d$ in the São Tomé's vitric Andosol (7.84%) is rather high, according to *Sanchez, Palm & Buol (2003)*. This soil presents moderate concentrations of non-crystalline Fe and Al, but a large amount of crystalline iron oxyhydroxides (Table 1). On the other hand, the silandic Andosol of Rwanda has a Fe$_o$/Fe$_d$ ratio of 0.75, indicating the predominance of non-crystalline forms of iron over crystalline forms. In the São Tomé vitric Andosol, the aluminium extractable by dithionite-citrate is higher than that extractable by acid oxalate, which indicates the presence of aluminium in the crystalline lattices of iron oxyhydroxides.

Values of pH in NaF higher than 9.5 indicate the presence of allophane and/or organo-aluminium complexes (*IUSS Working Group WRB, 2015*), materials active in the fixation of anions.

The soil samples were air-dried and sieved (<2 mm) prior to analysis.

## Sorption and desorption isotherms

To determine the sorption isotherms, 0.5 g of soil were equilibrated with 10 mL of solution of phosphate in 0.02 M NaNO$_3$ as background electrolyte at room temperature (293 K). P concentrations in equilibrating solutions ranged between 0 and 100 mg L$^{-1}$ (0, 2, 4, 10, 20, 40, 60 and 100 mg L$^{-1}$) and the solution pH was adjusted to the value of soil pH. The suspensions were shaken for 24 h, centrifuged at 6,000 rpm. for 15 min, and filtered through acid-washed filter paper. The determinations were carried out in triplicates. Phosphorus was determined in the equilibrium solution by visible spectrophotometry with molybdate by the ascorbic acid-reduced method (*Kuo, 1996*). Adsorbed phosphorus was calculated as the difference between added P and P in the equilibrium solution.

Desorption experiments were carried out in soil samples previously equilibrated with various phosphate concentrations. After removing the supernatant solution, 10 mL of 0.02 M NaNO$_3$ solution adjusted to the soil pH were added, the suspensions shaken for 24 h, centrifuged at 6,000 rpm. for 15 min, and filtered through acid-washed filter paper. Phosphorus was determined in the equilibrium solution as in the sorption experiments. The P concentration remaining in the adsorbed phase was calculated from the concentration adsorbed after the sorption equilibrium and the concentration released to the solution, making a correction to take into account the solution embedded in the solid after sorption equilibrium.
Sorption and desorption isotherms were obtained by plotting the phosphorus concentration in the adsorbed phase versus the solution concentration at equilibrium. The isotherms were fitted to mathematical models, namely the Langmuir, Freundlich and Temkin isotherms (*Mead, 1981*).

The Langmuir isotherm is described by the equation $Q = Q_{max} * K_L * c / (1 + K_L * c)$, where $Q$ is the concentration of the adsorbed anion (mmol kg$^{-1}$), $c$ is the concentration in the liquid phase at equilibrium ($\mu$mol L$^{-1}$), $K_L$ is a constant related to the adsorption energy and $Q_{max}$ (mmol kg$^{-1}$) is the maximum adsorption capacity. The Langmuir model assumes the adsorption of a monolayer, that all the adsorption sites have the same adsorption energy and that there is no interaction among the adsorbed molecules (or ions) (*Goldberg, 2005*).

The Freundlich isotherm is described by the equation $Q = K_F * c^n$, where $Q$ and $c$ have the same meaning as in the Langmuir equation, $K_F$ is the adsorption constant and $n$ is a constant whose value varies between 0 and 1. The Freundlich model assumes that the adsorption surface is heterogeneous and that the adsorption energy decreases exponentially as the concentration in the adsorbed phase increases. It does not predict an adsorption maximum (*Goldberg, 2005*).

The Temkin isotherm is described by the equation $Q = (RT/b) * \ln(Ac)$ or $Q = B * \ln(Ac)$, where $Q$ and $c$ have the same meaning as in the Langmuir equation, $R$ is the ideal gas constant (8.314 J mol$^{-1}$ K$^{-1}$), $T$ is the absolute temperature (in our case 293 K, that is 20 °C) and $A$ and $b$ are constants related to the heat of adsorption. It considers that the adsorption surface is heterogeneous and that the adsorption energy decreases linearly with the concentration in the adsorbed phase. There is no maximum adsorption (*Goldberg, 2005*).

## RESULTS

P sorption was very high in both the silandic and the vitric Andosols. In both soils, phosphorus sorption increased continuously as the phosphorus concentration in the equilibrating solution increased. P sorbed reached 47.7 $\mu$mol P g$^{-1}$ in the silandic Andosol (Rwanda) and 41.6 $\mu$mol P g$^{-1}$ in the vitric Andosol (São Tomé) for a P concentration of 3.2 mmol L$^{-1}$ (100 mg L$^{-1}$) in the equilibrating solution. The percent sorbed P values (Table 2) ranged from 77.0 to 98.1 in the silandic Andosol (Rwanda) and from 65.6 to 95.9 in the vitric Andosol (São Tomé). The highest percentages corresponded to 10–20 mg P L$^{-1}$ in the equilibrating solution (Fig. 1).

The sorption isotherms for both soils (Fig. 2) did not fit the Langmuir or Freundlich models. On the contrary, they fitted the Temkin model ($r^2 = 0.9702$ for the vitric Andosol and 0.9770 for the silandic Andosol). The fitted equations were $Q = 9.6843 \ln(0.2519\,c)$ for the silandic Andosol and $Q = 8.3025 \ln(0.1367\,c)$ for the vitric Andosol. Alternatively, the experimental points can be adjusted to two straight lines, the first with steep slope and the second with gentle slope:

$Q = 0.3792c$; $\quad Q = 0.0197c + 33.363$ for the silandic Andosol (Fig. 3)

$Q = 0.3268c$; $\quad Q = 0.0178c + 22.453$ for the vitric Andosol (Fig. 4).

**Table 2  Percent sorption and desorption in both studied soils.**

| P in the equilibrating solution (mg L$^{-1}$) | Vitric Andosol | | Silandic Andosol | |
|---|---|---|---|---|
| | % sorption | % desorption | % sorption | % desorption |
| 2 | 81.9 | 21.0 | 90.4 | 19.6 |
| 4 | 90.5 | 10.1 | 95.5 | 8.4 |
| 10 | 95.9 | 4.2 | 97.6 | 3.2 |
| 20 | 95.6 | 3.3 | 98.1 | 1.8 |
| 40 | 94.2 | 3.4 | 96.7 | 1.5 |
| 60 | 80.8 | 3.6 | 94.5 | 1.7 |
| 100 | 65.6 | 2.4 | 77.0 | 1.2 |

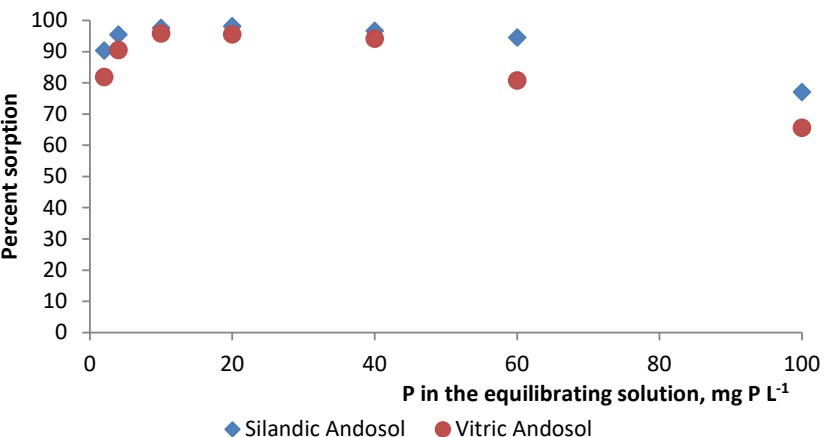

**Figure 1  Percentage of P sorption for various P concentrations in the equilibrating solution.**

The P desorption was always less than 21% of the P sorbed in both soils (Table 2). The desorption isotherms (Figs. 3 and 4) also fitted the Temkim model. The fitted equations were $Q = 22.014 \ln (0.1286 \, c)$, $r^2 = 0.9448$ for the silandic Andosol and $Q = 15.436 \ln (0.1034 \, c)$, $r^2 = 0.9855$ for the vitric Andosol.

## DISCUSSION

The concern raised by the sorption of phosphate by soils has a double aspect: agronomic and environmental. Phosphorus is an essential nutrient, so its availability in agricultural soils is crucial for plant development. On the other hand, the existence of excess phosphorus gives rise to its export to aquatic environments, where it can be the cause of eutrophication processes. In soils poor in available phosphorus, like those in the present study (Table 1), the main concern is the availability of this macronutrient, which can be reduced by its adsorption by soil components, in particular non-crystalline materials. A strong retention is an undesirable process, particularly if poorly reversible. On the contrary, in over-fertilized

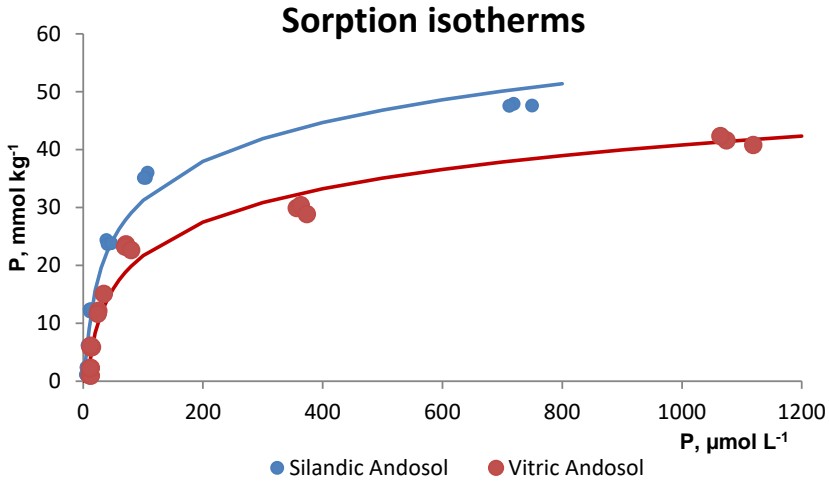

**Figure 2** Sorption isotherms fitted to the Temkin model.

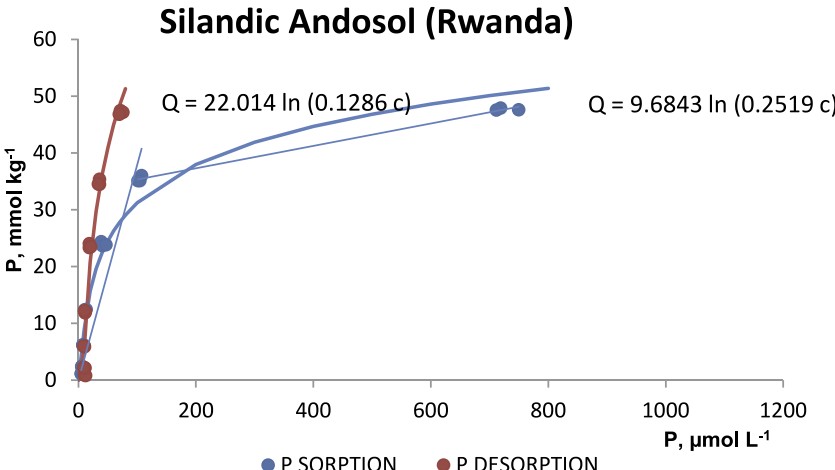

**Figure 3** Sorption and desorption isotherms for the silandic Andosol.

soils phosphorus sorption is a desirable process, as it protects aquatic environments from pollution.

The highest values of P sorption by the studied soils are comparable to the P sorption maxima reported for some volcanic soils in Indonesia (*Van Ranst et al., 2004*) or Azores, Portugal (*Auxtero, Madeira & Sousa, 2008*). The P sorption is somewhat higher in the silandic Andosol, richer in oxalate-extractable aluminium and iron (Table 1), compared to the vitric Andosol. The pH in NaF higher than 10 in the silandic Andosol indicates high phosphate fixation by allophane or imogolite (*Sanchez, Palm & Buol, 2003*). On the other hand, the remarkable P sorption by the vitric Andosol may be related to its richness in dithionite-extractable iron (Table 1), higher than the threshold value established by *Sanchez, Palm & Buol (2003)* for high phosphate fixation by iron and aluminium oxides.
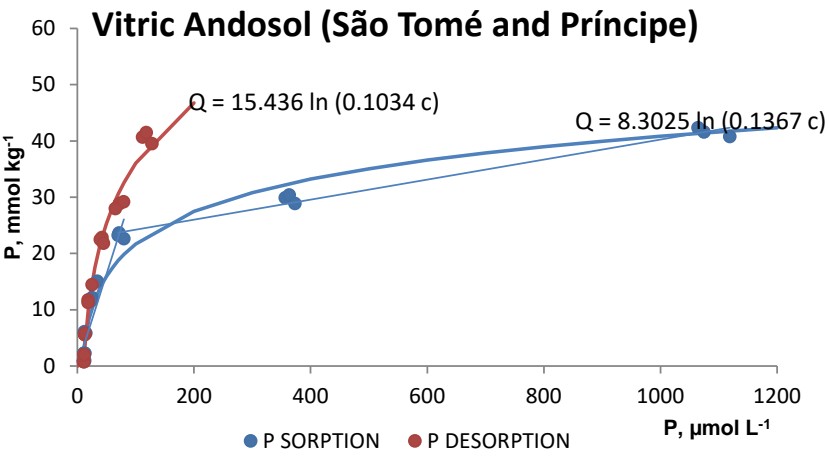

**Figure 4** Sorption and desorption isotherms for the vitric Andosol.

Several authors have reported a strong correlation between P sorption and Fe and Al extracted by acid oxalate (*Monterroso Martinez, Fernandez-Marcos & Alvarez Rodriguez, 1996*; *Van Ranst et al., 2004*), while others suggest that crystalline Fe is more important for P adsorption than amorphous Fe (*Auxtero, Madeira & Sousa, 2008*).

The P adsorptions at low P concentrations for the two Andosols were both higher than the values reported by *Duffera & Robarge (1999)* for an Ustivitrand (vitric Andosol) developed from volcanic ash in Ethiopia. It should be noted that the soil used by these researchers was formed in a climate dryer than that prevailing in the locations of the soils of the present study and had significantly lower concentrations of $Fe_d$ and $Al_d$ (1.6% and 0.15%, respectively). Furthermore, these authors studied a much lower range of concentrations than the one used in the present work.

By contrast, when comparisons are made with an Andosol developed from volcanic ash in Tanzania (*Gimsing, Szilas & Borggaard, 2007*), it is found that the values of P adsorption are quite similar. The similarity with the silandic Andosol is not surprising, since the $Al_o$ concentration of the Tanzanian soil and the Rwanda silandic Andosol are comparable, although the concentrations of $Fe_o$, $Al_d$ and $Fe_d$ of the Tanzanian soil were considerably lower than those of both Andosols in the present study.

Interestingly, the P adsorption by an Andosol of the Bambouto Mountains (West Cameroon), in the Cameroon Hot Line (*Siewe et al., 2008*), very close to the São Tomé Andosol, was very similar to those of the Andosols in the present study, with a maximum of 4.75 mg $PO_4$ $g^{-1}$ (50 mmol $kg^{-1}$).

The high P sorption, particularly in the silandic Andosol, indicates that this element will be retained to a large extent in a non-available form. In these soils, an efficient management of phosphate fertilization is crucial, with high applications of slow-release phosphorus fertilizers or repeated localised applications of small quantities of P fertilizers (*Arduino et al., 1993*). Organic substances, such as humic and fulvic acids or low molecular weight organic acids, compete with phosphate for adsorption sites (*Sibanda & Young,*

*1986*; *Antelo et al., 2007*); therefore, organic substances may significantly reduce phosphate adsorption and a sound management of organic matter and crop residues in phosphate adsorbing soils can help to improve phosphorus availability (*Zhu, Li & Whelan, 2018*).

Unlike the present study, in other research conducted for Andosols (*Van Ranst et al., 2004*; *Auxtero, Madeira & Sousa, 2008*) the best fit corresponded to Langmuir isotherms. Both the Langmuir and Freundlich models well describe P sorption on a variety of Australian soils (*Singh & Gilkes, 1991*), Thai Oxisols and Ultisols (*Wisawapipat et al., 2009*), soils developed from basic rocks in Portugal (*Horta et al., 2013*), and forest and vineyard soils from Galicia, Spain (*Romar-Gasalla et al., 2016*). The Langmuir, Freundlich and Temkin models can be used to describe P sorption on various Chinese soils (*Wang et al., 2013*). The Langmuir model described satisfactorily the P adsorption by four soils from Ethiopia, including an Andosol (*Duffera & Robarge, 1999*), as well as by four variable-charge tropical soils from Tanzania, including an Andosol developed from volcanic ash (*Gimsing, Szilas & Borggaard, 2007*). The phosphate adsorption by an Andosol of the Bambouto Mountains (West Cameroon) was also in agreement with the Langmuir model (*Siewe et al., 2008*).

The Langmuir equation is based on the assumption that the affinity of the surface for the adsorbate remains constant throughout the adsorption process. However, the adsorption of phosphate (and of other specifically adsorbed anions) increases the negative charge of the adsorbing surface, so that the adsorption of additional phosphate ions becomes increasingly difficult (*Barrow, 1978*). For this reason, the Langmuir equation only describes adsorption for low solution concentrations (*Barrow, 1978*; *Goldberg, 2005*). The Freundlich equation, which assumes that the affinity of the surface for the adsorbate decreases exponentially as the adsorption progresses, also describes well the adsorption at low solution concentrations (*Barrow, 1978*; *Goldberg, 2005*). The Temkin equation, which assumes that the affinity of the surface for the adsorbate decreases linearly as the adsorption progresses, is valid for an intermediate range of concentrations (*Goldberg, 2005*). The different ranges of concentration may explain the discrepancy between the present study and other similar research.

The fits in the present study are consistent with the specific adsorption of phosphate anion and indicate that the sorbing surface is heterogeneous. The fitted model cannot predict a limit to P sorption. The fit to two straight lines would indicate the presence of two types of sorbing sites: at low P concentrations in the equilibrating solution, phosphate would be sorbed onto high energy sites; at higher concentrations in solution, phosphate would be sorbed also onto lower energy sites.

The higher values of the B and A Temkin constants for the silandic Andosol compared to the vitric Andosol (9.6843 vs. 8.3025 and 0.2519 vs. 0.1367, respectively) indicate higher sorption energy and higher sorption equilibrium constant, respectively.

The silandic Andosol presents not only higher sorption, but also lower percentage of desorption (Table 2). That is, in the silandic Andosol phosphate is retained to a greater extent and is harder to desorb, which is in agreement with the higher sorption energy.

The sorption and desorption isotherms (Figs. 3 and 4) show a pronounced hysteresis: for a given value of the phosphorus concentration in the liquid phase, the concentration in the sorbed phase is greater during desorption. The irreversibility of phosphate sorption

is comparable to that reported by *Auxtero, Madeira & Sousa (2008)* for Azorean Andosols with high contents of allophane, $Al_o$, $Al_d$ and $Fe_d$.

## CONCLUSIONS

Both the silandic Andosol of Rwanda and the vitric Andosol of São Tomé showed high phosphorus sorption, being the silandic Andosol the one which sorbs more and desorbs less. The sorption by the silandic Andosol is related to active aluminium and iron species, while crystalline iron and aluminium oxides are mainly responsible for sorption in the vitric Andosol.

The strong phosphate sorption by the studied soils is a serious limiting factor for plant growth, which requires a careful management of phosphorus fertilization, including an appropriate management of organic matter.

The low levels of desorption in both soils indicate a low risk of phosphorus leaching losses.

### Funding

The authors received no funding for this work.

### Competing Interests

Maria Luisa Fernandez-Marcos is an Academic Editor for PeerJ.

### Author Contributions

- Sara Gonzalez-Rodriguez performed the experiments, analyzed the data, prepared figures and/or tables, authored or reviewed drafts of the paper, approved the final draft.
- Maria Luisa Fernandez-Marcos conceived and designed the experiments, analyzed the data, prepared figures and/or tables, authored or reviewed drafts of the paper, approved the final draft.

### Data Availability

The raw data are available in Data S1.

### Supplemental Information

Supplemental information for this article can be found online at http://dx.doi.org/10.7717/peerj.5820#supplemental-information.

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
