# Peer review of "Phosphate sorption and desorption by two contrasting volcanic soils of equatorial Africa"

_PeerJ, doi:10.7717/peerj.5820_

## Round 0.1 · original submission · Major Revisions

Please find the reviewers comments. They are pertinent to the manuscript. I agree with the third reviewer' comments that the current manuscript needs some major improvements before acceptance.

Reviewer 1 ·

Basic reporting

The merit of this study is vague as comparable adsorption extents and mechanisms have been determined for decades.

Experimental design

Lack of sufficient details for methods.

Validity of the findings

The current data are not sufficient to describe the comprehensive adsorption and desorption behaviors.

Additional comments

Manuscript Number: 27906
Title: Phosphate sorption and desorption by two volcanic soils of equatorial Africa

This study aimed to determine the adsorption and desorption of phosphate on two Andosol soils in equatorial Africa using the sorption isotherms. Phosphate adsorption on agricultural soils and soil minerals have been widely studies for decades. The sorption capacities and mechanisms for phosphate on Andosols reported in this study are comparable to that of previously studies. Although impact and novelty are not assessed in PeerJ, this journal asked research question to fill an identified knowledge gap. Given that the mechanisms to control the phosphate adsorption on volcanic soils have been well developed, authors failed to convince the merit of this study. The only difference between this and previous studies is that the sorption isotherms in this study could not be fitted using the Langmuir or Freundlich equations appropriately, which may be caused by the various sorption mechanisms. Thereby, I would suggest authors to discuss the possible mechanisms that could account for the inappropriate fitting using the Langmuir or Freundlich equations. In addition, I suggest to add the adsorption kinetics for phosphate adsorption on Andosol to provide a more comprehensive sorption mechanisms. For all these reasons, I recommend a major revision for this manuscript.

Specific comments:
Line 106-108: “The concentration of Fed is very high in the São Tomé's vitric Andosol, which presents moderate concentrations of non-crystalline Fe and Al, but a large amount of crystalline iron oxyhydroxides.” Please show the exact numbers for all results to make this description more concise.

Line 110-112: the aluminium extractable by dithionite-citrate is higher than that extractable by acid oxalate, which indicates the presence of aluminium in the crystalline lattices of “iron” oxyhydroxides. Is the “iron” in the sentence a typo?

Sorption and desorption isotherms:
What is the tested pH? How long did the equilibration take? Did the pH remained constant over the course of equilibration? How to control the pH? Please provide more details for the methods.

Line 138-154: “The Langmuir model assumes the adsorption of a monolayer, that all the adsorption sites have the same adsorption energy and that there is no interaction among the adsorbed molecules (or ions).” Add references to the description of models for adsorption isotherms.

Line 156: “P sorption was very high in both the silandic and the vitric Andosols.”
Please provide the numbers for the P sorption capacity. Are the P sorption capacities of silandic and vitric Andisols generally higher than other soil or mineral samples?

Line 156-162 and Figure 1: This paragraph focused on the proportion of P adsorbed by soil samples decreased with the increasing initial addition. However, it is a very well-known and common sense for the adsorption behavior. Such description seems redundant. What is the reason to address this point specifically?

Line 163: Langmuir or Freundlich equations are two most general models to fit the data of adsorption isotherms. Why couldn’t these two models fit the P adsorption data in this study? What mechanisms were involved in the P adsorption on Andosols, resulting in the in appropriate fitting using the Langmuir or Freundlich equations? This information is the key contribution for this study.

Line 177-185: The information in this paragraph is repetition of the content in the information.

Line 186-187: The highest values of P sorption derived in this study are comparable to that for other volcanic soils. Given that the extents and mechanisms of P adsorption in this study are similar to that widely reported previously, what is the contribution of this study?

Line 207-209: Authors proposed that the fits for P adsorption isotherms indicated that the sorbing surface is heterogeneous and that there is no sorption maximum, i.e. there is no apparent limit to P sorption. Is the deduction between no fitted sorption maximum and no apparent limit to P sorption reasonable?

Line 209-212: The proposed sorption mechanisms are vague. What do the “high energy sites” and “lower energy sites” mean by? Please add specific sorption mechanisms in the manuscript.

·

Basic reporting

no comment

Experimental design

no comment

Validity of the findings

no comment

Additional comments

Abstract
Line 14 -- replacing “and” with “but” could be better.
Line 16-17 -- what is attributable to active species of aluminum and iron? “The major limiting factor” or “phosphate fixation”?
Introduction
Line 77 -- “Cameroon Hot Line” is a proper noun, which means “h” and “l” in the manuscript should be in upper case.
Line 79-80 -- What do you want to achieve by studying the adsorption and desorption of phosphate by surface horizons of agricultural soils in equatorial Africa? Also, in line 84-86, the reason why you did this research may not be adequate. I suggest you rewrite the reason why you did this research and the purpose of doing this research to make future readers better understand the necessity of the study and the motive for this paper. By the way, I think you may cite papers on volcanic soils on Africa relevant to your topic if you can so that future readers may better know the scarcity of research concerned on volcanic soils in Africa.
Results
Line 171 -- It should be “was” instead of “is”.
Discussion
Line 200 -- “slow-release phosphorus fertilizers” may be better.
Line 202 -- There should be a comma before “the best fit”.
Line 203 -- It should be “well describe”
Line 216 -- The word “presents” should be before the word “not only”.
Line 222-223, “o” and “d” should be in subscript.
Reference
The title of papers and books cited should be basically in lower case.
Figure
In figure 1, using “Percentage of” is better.
You can also show the equation of model in Fig. 3 and 4 to make it easier to understand.
Table
In table 1, “o” and “d” should be in subscript.

Reviewer 3 ·

Basic reporting

The work presented is a very basic study of the sorption-desorption process of phosphate in volcanic soils. On the other hand authors worked only with Andisols, therefore the title is not appropriate. In the literature there are many antecedents referred to other orders of volcanic soils. Despite the above, the authors do not adequately justify the need to carry out a study of the type presented. The most essential question would be if the soils studied are significantly different in their properties from the numerous other soils (Andisols) in which phosphate adsorption has been studied.

Experimental design

In the experimental design authors considered only two soils for the study, so conclusions obtained are very limited. The characterization of soils and the sorption and desorption experiments were performed properly. However, isotherms must be carried out at constant and controlled temperature. Methods used were described with sufficient detail and can be replicated.There are other studies that could be made for obtaining more antecedents, for example the study of pH and temperature effects are very important to characterize both processes and discuss results more properly.

Validity of the findings

Comments are in the general comments for the author.

Additional comments

In the manuscript “Phosphate sorption and desorption by two volcanic soils of equatorial Africa” the study is performend by using the surface horizons of two agricultural soils of Africa. The analysis of data is based on the different composition of both soils related to the more important presence of amorphous Fe-and Al active forms in a silandic Andosol and the high amounts of crystalline iron and aluminum oxides in a vitric Andosol. The work is well presented but it is not novelty because the subject has already been investigated in volcanic derived soils and specifically in allophanic and non-allophanic Andisols. This aspect could be discussed for example in the introduction section. There are important references that could be considered for example: Path Analysis of Phosphorus Retention Capacity in Allophanic and Non-allophanic Andisols (Hashimoto, Y; Kang, J.; Matsuyama, N; et ál. Soil Sci. Soc. Am. J., 76 (2) 441-448 (2012), Spatial distribution assessment of extractable Al, (NaF) pH and phosphate retention as tests to differentiate among volcanic soils (Valle, S; Carrasco, G., Pinochet, D. et al. CATENA, 127, 17-25, 2015), Phosphorus sorption maxima and desorbability in selected soils with andic properties from the Azores, Portugal (Auxtero, E.; Madeira, M.; Sousa, E., Geoderma 144 (2008) 535–544. As presented and with the work carried out, the impact and novelty would be relevant only for a local level interest. As can be observed from these manuscripts and also in several works cited by authors, the antecedents correspond to a great number of soils, for which an extensive interpretation of data has been performed by taking into account a great variety of soil properties. In the present work the results and discussion are restricted to only two soils and no interpretation could be made, apart from the one presented, where the other properties of the soils are included. According to the above considerations conclusions are too general and limited to a very scarce supporting results.

---

## Round 0.2 · accepted · Accept

Thank you for considering your paper in PeerJ. I am happy with your revisions, and so I am writing to inform you that your manuscript - Phosphate sorption and desorption by two contrasting volcanic soils of equatorial Africa - has been Accepted for publication. Congratulations!